# Exploring cross-tissue DNA methylation patterns: blood–brain CpGs as potential neurodegenerative disease biomarkers
Vanessa Mendonça [1,2], Sheila Coelho Soares-Lima[3] & Miguel Angelo Martins Moreira [2] ✉

The difficulty of obtaining samples from certain human tissues has led to efforts to find accessible sources to analyze molecular markers derived from DNA. In this study, we look for DNA methylation patterns in blood samples and its association with the brain methylation pattern in neurodegenerative disorders. Using data from methylation databases, we selected 18,293 CpGs presenting correlated methylation levels between blood and brain (bb-CpGs) and compare their methylation level between blood samples from patients with neurodegenerative diseases (Alzheimer's disease, Parkinson's disease, Multiple Sclerosis, and X Fragile Syndrome) and healthy controls. Sixty-four bb-CpGs presented significant distinct methylation levels in patients, being: nine for Alzheimer's disease, nine for Parkinson's disease, 28 for Multiple Sclerosis, and 18 for Fragile X Syndrome. Similar differences in methylation pattern for the nine Alzheimer's bb-CpGs was also observed when comparing brain tissue from patients vs. controls. The genomic regions of some of these 64 bb-CpGs are placed close to or inside genes previously associated with the respective condition. Our findings support the rationale of using blood DNA as a surrogate of brain tissue to analyze changes in CpG methylation level in patients with neurodegenerative diseases, opening the possibility for characterizing new biomarkers.

The difficulty of obtaining certain human tissues has led to efforts to find epigenetic signatures in more accessible samples as DNA from cells of the oral mucosa, cervical cells, or peripheral blood[1–8]. Lymphocyte DNA methylation has been widely used as a tool for aiding the diagnosis and prognosis of various diseases, for example: in determining the risk of breast cancer[9–13], development of osteoporosis in postmenopausal women[14,15] and in determining the response to medications in patients with type 2 diabetes mellitus[16]. The association between methylation in tissues from difficult-to-obtain organs and blood is particularly important to neurologic and psychiatric disorders due to risks and difficulties to access brain tissues. For instance, it has been relevant in the search for new DNA methylation biomarkers for early diagnosis without the need to access a brain tissue sample[17,18].

The use of blood cells as a surrogate for some neurodegenerative diseases can reflect the molecular and cellular changes involving various immunological mediators, including T lymphocytes. This type of cell accounts for part of nucleated blood cells and can cross the blood-cerebrospinal fluid barrier and the blood-brain barrier, circulating and being present in both brain tissue and blood[19]. The cross of these barriers is supported by the detection of T lymphocytes in brain tissue of Alzheimer's[20], Parkinson[21,22], and Multiple Sclerosis[23] diseases. Since DNA methylation can be influenced by the cell environment, we hypothesized that blood cells can acquire and maintain specific CpG methylation patterns after circulating in the brain and upon its return to the bloodstream.

In this study, we looked for CpGs for which the level of methylation in blood DNA is correlated with the level observed in brain tissues in patients with neurodegenerative disorders. Our findings show that there is a distinct methylation profile for specific CpGs in individuals affected by Alzheimer's, Parkinson's, Multiple Sclerosis, and Fragile X Syndrome when compared with healthy subjects.

## Material and methods

### Selection of CpG sites with correlated DNA methylation pattern between peripheral blood and brain

Two databases, IMAGE-CpG[3] and BECon[4], were used to establish CpGs where methylation in blood and brain tissue was correlated. These datasets were generated using Infinium HumanMethylation450K and Infinium EPIC (850 K) assays and comparing methylation data from (i) brain

[1]Genetic Graduation Program, Genetics Deparment, Universidade Federal do Rio de Janeiro, Rio de Janeiro, Brazil. [2]Tumoral Genetics and Virology Program, Instituto Nacional de Cancer, Rio de Janeiro, Brazil. [3]Molecular Carcinogenesis Program, Instituto Nacional de Cancer, Rio de Janeiro, Brazil. ✉e-mail: miguelm@inca.gov.br

resection samples, blood, saliva, and oral tissue samples from patients with refractory epilepsy and (ii) post-mortem blood and brain tissue samples from individuals without any psychiatric diagnosis or history of substance abuse. A total of 18,293 CpGs indicated by at least one database with a correlation coefficient > |0.70| were selected, respecting the statistical analysis used by each of them (Pearson correlation in IMAGE-CpG and Spearman in BECon; Supplementary Data 1).

### Establishment of the methylation profile in samples from individuals with neurodegenerative diseases

To establish the model, we compared the beta-values for the 18,293 CpGs between patients and healthy controls. The beta-values, which represent the proportion of methylated CpG, were estimated by the ratio of methylated/(methylated+unmethylated) signals and were obtained from patients with neurodegenerative diseases in the Gene Expression Omnibus (GEO) database. In this study we adopted an exploratory approach, respecting the normalization and patient selection applied by the authors of each dataset, without specific sub-division based on age, sex, ethnicity, or disease stage. Only studies that evaluated the methylation profile using the Infinium Human Methylation 450 K or Infinium Methylation EPIC arrays were included, as these are the methodologies used in IMAGE-CpG and BECon.

### Statistics and reproducibility

The study comprised six different datasets containing beta values (Supplementary Table 1). Two for Alzheimer's disease: GSE153712, comprising 161 patients and 471 controls, for blood samples; and GSE72778, comprising 125 samples from 18 Alzheimer's patients and 135 control samples from 21 controls, for brain tissues. Two for Parkinson's disease: GSE111629, including methylation data from blood DNA of 335 patients and 237 controls; and GSE195834, containing data from the brain DNA of 38 patients and 40 controls. One for Fragile X syndrome: GSE41273, containing blood DNA data from 62 males, including 9 patients and 53 controls. One for multiple sclerosis: GSE106648, containing beta values of methylation for blood DNA from 140 patients and 139 controls.

## Results

### Selection of CpG presenting correlation of DNA methylation levels between peripheral blood and brain

To identify the CpGs presenting correlated methylation levels between blood and brain tissues in paired samples, we analyzed the databases IMAGECpG and BeCon. First of all, 47,531 CpGs presented methylation levels with correlation coefficient $\geq |0.70|$, when comparing blood and brain-paired samples, being: 47,360 from the IMAGECpG dataset, and 1131 from the BeCon dataset. IMAGECpG utilized two methylation arrays (HumanMethylation 450 K and Infinium EPIC 850 K), and CpGs present in both arrays exhibiting inconsistent results regarding the correlation between blood and brain were excluded. We identified 11,772 CpGs exclusive to the Infinium EPIC 850 K array (considered the most comprehensive), and an additional 5390 CpGs shared between the HumanMethylation450K and Infinium EPIC 850 K arrays and with concordant results in both analyses. Furthermore, we included the 1131 CpGs from the Becon dataset, resulting in a total of 18,293 CpGs (Fig. 1 and Supplementary Data 1). These CpGs are hereafter referred to as bb-CpGs, from blood and brain correlated CpGs.

A schematic representation of the regions where the CpGs are placed, according to their position concerning CpG islands or genes, is shown in Fig. 2, as well as the nomenclature used to refer to each of these positions.

### Establishment and validation of the blood-brain correlation model

To identify sites with differentially methylated levels in blood and to verify if the same profile could be found in the brain, the methylation of the bb-CpGs was analyzed in two case/control datasets of Alzheimer's disease: GSE153712, for blood, and GSE72778, for brain. A total of nine bb-CpGs

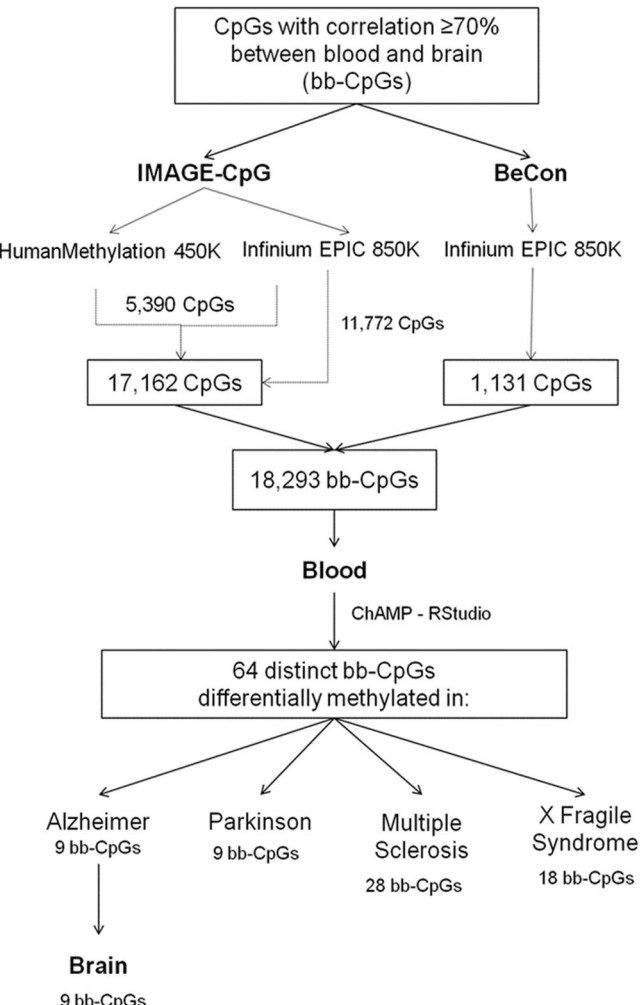

**Fig. 1 | Selection of 18,293 presenting correlation of DNA methylation levels between peripheral blood and brain (bb-CpGs).** Based on filtering CpGs from IMAGE-CpG and BECon database.

were found to be differentially methylated, being all hypermethylated in blood samples of Alzheimer's Disease (AD) patients in comparison to healthy controls (Fig. 3a). The chromosomes, genes, and locations of these bb-CpGs in respect to gene structure and CpG islands are shown in Supplementary Table 2.

To evaluate if these nine bb-CpGs present a similar differential methylation profile between brain samples from AD patients and healthy controls, we used the GSE72778 dataset. All brain DNA samples were grouped for disease or control group, and tissue samples from different brain regions were analyzed together. The results showed that all nine bb-CpGs were also significantly hypermethylated in brain tissue of AD patients, similar to what was found in the analysis with DNA derived from blood samples (Fig. 3b).

### Other neurodegenerative diseases

Using the same set of 18,293 CpGs we carried out similar analyses for other neurodegenerative conditions (Parkinson's disease, Multiple Sclerosis, and X Fragile Syndrome), comparing the methylation levels of DNA from blood between affected individuals and healthy controls. Cases and controls were specific to each dataset.

The analyses carried out in patients with Parkinson's disease (dataset GSE111629) identified nine differentially methylated bb-CpGs. These sites were found to have a significant decrease in methylation in the Parkinson's disease group compared to the control group (Supplementary Fig. 1). The

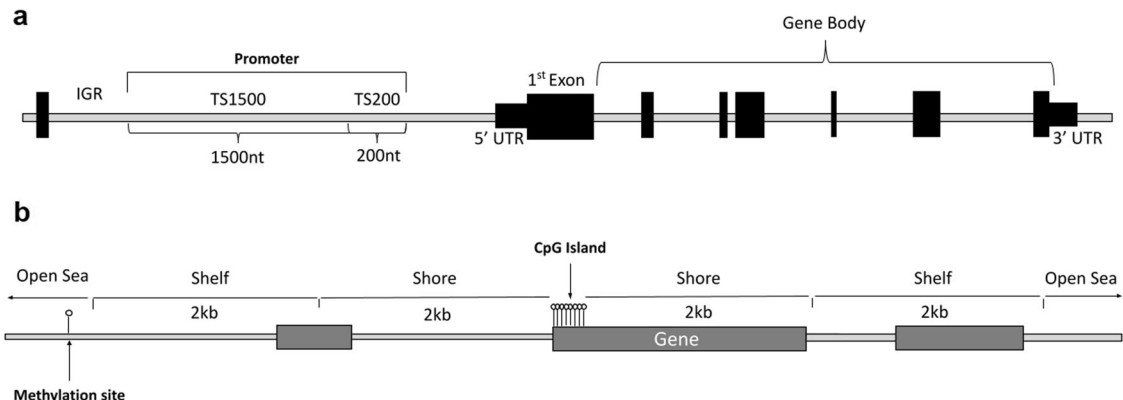

**Fig. 2 | Distribution of CpGs according to their relative position. a** In respecto to CpG islands or **b** in respect to genes and the nomenclature used to refer to each of these positions.

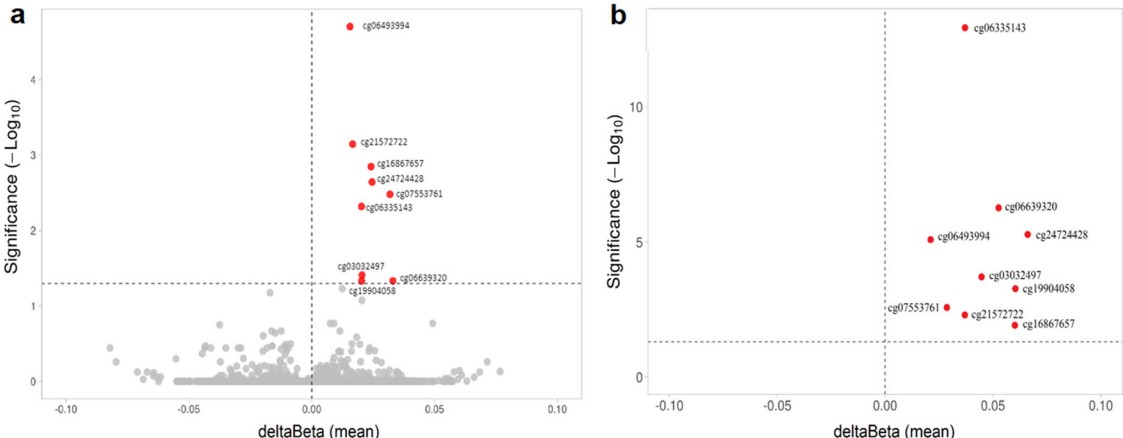

**Fig. 3 | Difference in average beta values between AD patients and controls for each CpG analyzed.** The y-axis shows the adjusted *p*-values by Benjamini-Hochberg on a logarithmic scale (-Log10(p)). The trace line represents the threshold for significant values (≥1.30). On the x-axis is the difference in average beta values (delta beta) between cases and controls, being positive when there is higher methylation of the CpG in the case group and negative when there is less methylation in the same group. **a** Comparison between DNA samples derived from blood samples from AD patients and health controls (GEO ID: GSE153712). **b** Comparison between 125 DNA samples derived from different brain regions of 18 AD patients and 135 from 21 health controls (GEO ID: GSE72778).

nine bb-CpGs were mapped to eight different chromosomes and the genes associated with them, as well as the position of each bb-CpG, are detailed in Supplementary Table 3.

To carry out comparisons considering these nine bb-CpGs in brain tissues, the single data set meeting our criteria (i.e., analyzed by Human-Methylation 450 K or Infinium EPIC 850 K and containing patients and controls in the same dataset) was GSE195834, based on samples of primary motor cortex. Of the nine CpGs, eight presented the hypomethylated pattern found in blood samples, although only for one CpG the difference was statistically significant (cg12230709; Supplementary Fig. 2).

Considering Multiple Sclerosis (dataset GSE106648), 28 bb-CpGs were found to be differentially methylated between patients and healthy controls, with 12 significantly hypomethylated and 16 hypermethylated with respect to controls (Supplementary Fig. 3). The 28 bb-CpGs were distributed across nine different chromosomes, and most bb-CpGs were in HLA (Human Leukocyte Antigen) loci, in chromosome six, where genes associated with the disease are located. Information about differentially methylated bb-CpGs is presented in Supplementary Table 4.

In the analysis of the X Fragile Syndrome (dataset GSE41273), 18 bb-CpGs were found to be differentially methylated between patients and controls, with 14 sites significantly hypermethylated and four hypomethylated in patients (Supplementary Fig. 4). These 18 bb-CpGs were distributed across four different chromosomes, predominantly in the X chromosome and in the *ASFMR1* (antisense *FMR1*) gene, which has been described as associated with clinical phenotypes of FMR1-related disorders such as X Fragile Syndrome. The genes in which this and other differentially methylated bb-CpGs were found, as well as the location of each bb-CpG site with respect to the gene region, are shown in Supplementary Table 5.

The lack of CpG methylation data from brain tissues for Multiple Sclerosis and X-Fragile patients did not allow to verify if a similar trend to blood samples can be observed in brain for these diseases.

## Discussion

This study provided evidence for the association between the methylation levels of nine specific bb-CpGs from genomic DNA obtained from blood samples and the Alzheimer's disease. Supporting these findings, we confirmed the association of hypermethylation in these nine bb-CpGs in an independent dataset by comparing the methylation pattern in brain tissue of Alzheimer's patients with control individuals. Additionally, we found that six of these hypermethylated bb-CpGs were mapped to in promoter regions of genes previously associated with Alzheimer's, suggesting that the hypermethylation could be associated with a downregulation of these genes. This rationale is supported for *SCGN*, *ELOVL2*, *TRIM59*, and *FHL*. The *SCGN* gene, where cg06493994 is located and hypermethylated in patients, has been found to have a significant reduction in expression in the brain of Alzheimer's mouse models[24,25]. Neurons expressing SCGN are resistant to cell death in neurodegenerative Alzheimer's brain and its expression has been associated to reduced neurodegeneration and to

a possible protective effect against the disease[24,26]. The same could be applied to bb-CpGs cg21572722, cg16867657, and cg24724428 located in *ELOVL2*, a gene associated with increased risk of AD, for which hypermethylation was observed in the hippocampus of individuals with the disease[27,28]. The gene *TRIM59*, where cg07553761 is located, was associated with cell death signaling in the familial form of Alzheimer's disease when hypermethylated[29]. Finally, the *FHL2* (where cg06639320 is mapped to) was described as potentially associated with Alzheimer's disease, with its deficiency leading to neuronal migration delay and premature astrocyte differentiation[30].

The differentially methylated bb-CpGs in the blood of Parkinson's patients were mapped to genes associated with susceptibility or mechanistically associated with disease characteristics: *NQO2, RAB7A, TBC1D16, ULK1, CD302, CALD1* and *PRTN3*. The *NQO2* gene was suggested to be an important factor in Parkinson's disease, and promoter polymorphisms were associated with disease susceptibility[31,32]. Similarly, hypomethylation of the 5'UTR region of *RAB7A* could indicate lower expression of this gene, decreasing the degradation of α-synuclein aggregates[33]. The gene *TBC1D16* promotes the activity of Rab5C GTPase, which is implicated in Parkinson's disease[34–37]. ULK1-mediated autophagy induction occurs early during axonal degeneration[38] and ULK1 inhibition is responsible for causing protective effects on axonal degeneration and against neurodegeneration in the mouse model of Parkinson's disease[39]. The transcription of the genes *CD302* and *CALD1* were found to be downregulated in prefrontal cortex of Parkinson patients when compared with healthy individuals[40]. With respect to *PRTN3*, the literature shows no evidence of association with the disease characteristics.

We investigated these same nine CpGs in a single dataset available (GSE195834) containing beta-values derived from brain tissue of patients with Parkinson's disease. Of the nine CpGs, eight exhibited the hypomethylated pattern observed in blood samples, although only one CpG showed statistically significant differences (cg12230709; Supplementary Fig. 2). The limited brain representativeness, due to the sampling being restricted to the primary motor cortex, could account for the nonsignificant differences in the remaining CpGs and the divergent pattern of one of them.

For Multiple Sclerosis and Fragile X Syndrome, most bb-CpGs with differential methylation were clearly associated with the phenotypic presentation. For Multiple Sclerosis, an autoimmune condition, differential methylation in multiple *HLA-DR* loci in the blood reflects this fact, being corroborated by the increased expression of *HLA-DRB1* and *HLA-DRB5* in the brain of Multiple Sclerosis patients[41]. For Fragile X Syndrome, multiple bb-CpGs in the X-chromosome were found to be differentially methylated. Beyond the *ASFMR1* gene, placed head-to-head with *FMR1* gene (Fragile X Messenger Ribonucleoprotein 1 gene) in the X chromosome, two other genes, *THSD1P* (TSS) and *CTAG2* (exon 1), exhibited hypomethylation of bb-CpGs in promoter regions, which may correspond to increased protein expression. However, no previous data linked these two genes to Fragile X Syndrome. In spite of the association between Fragile-X Syndrome and Amyloid Beta Precursor Protein expression[42–44], no bb-CpG with differential methylation found in AD patients was differentially methylated in Fragile-X patients.

Interestingly, bb-CpGs found in intergenic regions (IGR) in Alzheimer's and Parkinson's dataset had at least one of the flanking genes previously associated with neurodegenerative diseases or neuronal development. For Alzheimer's disease, the single differentially methylated IGR bb-CpG (cg03032497) in blood samples was mapped in an enhancer region between the *SALRNA1* and *SIX1* genes. *SIX1* has been shown to participate in neuronal development[45] and has also been found to have differential expression in the superior temporal gyrus of Alzheimer's patients[46]; on the other hand, *SALRNA1* does not have a well-established relationship with Alzheimer's disease. The differentially methylated bb-CpG cg11725581 in Parkinson's Disease dataset is mapped between the *AIMP2* and *USP42* genes. *AIMP2* has a strong association with the brain features of Parkinson's disease and is responsible for increasing the

accumulation of α-synuclein[47]. On the other hand, the association between *USP42* and the disease is still unknown. For multiple Sclerosis, two of the three bb-CpGs found in IGR were placed in the HLA region, and the other (cg26328180) placed between genes (*LOC107985911* and *KMT2CP4*) not related with the disease's pathogenesis. In fragile X syndrome, the two bb-CpGs found in IGRs (cg27524192 and cg02180907, mapped between *RNA5S13* and *RNA5S14* genes, and *ERICH1* and *LOC286083*, respectively) had flaking genes not associated with the syndrome. However, the absence of CpG methylation data from brain tissues for Multiple Sclerosis and X-Fragile patients do not allow to confirm the same trend observed in blood samples, particularly for the genes not placed in HLA/MHC loci or X-chromosome.

No differentially methylated bb-CpGs nor the genes mentioned above were consistently identified across the diseases, suggesting that methylation patterns observed are specific for each neurodegenerative condition. On the other hand, the results found for Alzheimer's and Parkinson's Diseases in blood samples and brain tissues reinforce that the evaluation of methylation profile of bb-CpGs in blood can reflect epigenetic changes in brain tissue. Although analyzing DNA methylation in blood as a surrogate for brain has its limitations with regard to underrepresentation of the alterations that take place in the disease-affected tissue, it has a potential use in the identification of biomarkers that could help diagnosis and disease monitoring. In this context, the data presented here support the rationale of using blood as a surrogate of brain tissue to analyze changes in methylation profile in patients with neurological disease, opening the possibility for the characterization of new biomarkers.

## Reporting summary

Further information on research design is available in the Nature Portfolio Reporting Summary linked to this article.

## Data availability

The neurodegenerative diseases used for comparison were defined based on the availability of blood datasets at GEO database. (Supplementary Table 1) containing both patients and healthy individuals and included: Alzheimer's (GSE153712), Parkinson's (GSE111629), Multiple Sclerosis (GSE106648), and Fragile X Syndrome (GSE41273). The inclusion of Fragile X Syndrome is due to the association between Fragile X Mental Retardation Protein (FMRP) pathway and the translation of Amyloid Beta Precursor Protein[42–44], whose accumulation in the brain is a hallmark of Alzheimer's Disease. Additionally, we were able to investigate in brain tissue the blood differentially methylated CpGs corresponding to two of the analyzed diseases: Alzheimer's and Parkinson's. For Alzheimer's, we utilized a dataset with beta-values (GSE72778) obtained from DNA samples representing various brain regions (frontal lobe, parietal lobe, occipital lobe, caudate nucleus, cingulate gyrus, cerebellum, hippocampus, etc.) from patients and healthy controls. The Parkinson's dataset (GSE195834) was limited, focusing solely on DNA samples from the primary motor cortex of patients and unaffected controls.

## Code availability

Datasets containing normalized methylation beta values were loaded into RStudio and filtered to include only regions of interest using the list of 18,293 pre-selected CpGs. Then, we used a linear Bayesian regression model, implemented through the "champ.DMP()" function from the "ChAMP" package, to define differentially methylated CpGs (DMPs) between cases and controls of each disease data set. The DMPs were considered statistically significant if their adjusted $p$-values were ≤0.05.; Adjusted $p$-values for multiple comparisons were obtained using the Benjamini-Hochberg method considering the 18,293 CpGs. The volcano plots were generated using the data shown in Supplementary Data 2 with VolcaNoseR, (available at https://huygens.science.uva.nl/VolcaNoseR/).

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

## Acknowledgements

This work was supported by Brazilian Research Council (CNPQ, grant: 304833/2022-3), Carlos Chagas Filho Research Support Foundation of the State of Rio de Janeiro (FAPERJ, grant: 200.928/2021 and E-26/211.309/2021), Coordination of Superior Level Staff Improvement (CAPES, Brazil), Ministry of Health (Brazil), and Brazilian National Cancer Institute (INCA-Brazil).

## Author contributions

V.M., S.C.S.-L., and M.A.M.M. contributed with study design. V.M. and S.C.S.-L. carried out data analyses. V.M., S.C.S.-L., and M.A.M.M. contributed with manuscript writing and revision of the final version.

## Competing interests

The authors declare no competing interests.
