## [Peer Review File · Communications Biology]

Reviewers' comments:

Reviewer #1 (Withdrawn)

Reviewer #2 (Remarks to the Author):

Mendonça et al. have investigated the potential use of blood-brain CpGs as biomarkers in neurodegenerative disease. This is an interesting topic, and the overall design of the study is sound. Demographics and description of the datasets used is well described in tables, and figures provided are useful in interpretation of study design and results. In general the authors have clearly explained the use of different datasets/types to investigate the research question, but there are points when it is difficult to follow. My suggestions for improvement are provided below:

1) It would be beneficial for the authors to make clearer the use of blood/brain datasets and the diseases they correspond to, and to justify the use of these datasets.

2) There are many available brain DNA methylation datasets that correspond to different neurodegenerative diseases. It might be beneficial to compare results to a wider range of diseases/those with similar pathology. As the manuscript is reasonably short, there is room to expand the analysis. The datasets used for the Alzheimer's brain cohort are relatively old, it would be beneficial to look at DMPs from more recent studies.

3) There are several grammar/spelling mistakes, it would be beneficial to review the manuscript to identify and rectify such instances.

4) In results section 1 (Selection of CpG presenting correlation of DNA methylation levels between peripheral blood and brain), clarification on the numbers of CpGs identified would be beneficial. It is confusing to the reader about where the different number of CpGs referred to come from.

5) Results section 2 (Establishment and validation of the blood-brain correlation model) is unclear. Consider rewording the results to explain more clearly what the purpose of the use of the different datasets in this section was.

6) It should be acknowledged that there are limitations to the use of blood data as a surrogate, given that the number of CpGs with blood brain correlation is relatively small.

Reviewer #3 (Remarks to the Author):

Mendonça and co-workers selected DNA methylation (DNAm) sites showing high correlation between blood and brain tissue based on previously published datasets. These selected DNAm sites were tested for associations with several neurodegenerative diseases. In total, they identified 64 DNAm-disease associations. Medonça and co-workers conclude that these findings support the rationale for using blood as a surrogate tissue for brain-related phenotypes.

My main concerns are about the validity of the DNAm association analysis. I miss a clear description on pre-processing of the DNAm data and the linear regression models used for determining the associations.

Major comments

Introduction

1. You mention that DNAm from surrogate tissues (i.e., lymphocytes) are used to inform on diagnoses and prognoses of various diseases. The diseases you mention are all peripheral diseases, and one could argue that these peripheral diseases have widespread effects on the peripheral epigenome through e.g. the immune system. Your next conclusion is the importance to study peripheral DNAm in brain-related disorders. Could you provide a better rational why you think that peripheral DNAm might be informative for the brain, especially given the presence of the blood-brain barrier (BBB)?
2. The introduction lacks a clear rational why it is important to study the epigenome in neurodegenerative disorders. Could you expand on the study rational? Could you also expand on why you specifically study AD, PD, MS, and Fragile X?
3. *"The use of blood cells as a surrogate for some neurodegenerative diseases can reflect the molecular and cellular changes involving various immunological mediators, including T lymphocytes."* Could you provide references for these statements?
4. *"This type of cell accounts for about ~70% of nucleated blood cells and can cross the blood-cerebrospinal fluid barrier and the blood-brain barrier, circulating and being present in both brain tissue and blood (YAMAZAKI; KANEKIYO, 2017)."* The study cited here, discussed the possibility of a leaky BBB in Alzheimer's disease specifically. How does this translate to the other diseases you investigate in your manuscript? Can you assume this is the case for every neurodegenerative disorder, and why?

Methods

1. *"The CpGs indicated by at least one database with a positive or negative correlation above 70% were selected, respecting the statistical analysis used by each of them."* Could you clarify whether you are using the correlation coefficient of $|0.7|$, or the explained variance of 70%?
2. How did you deal with the DNAm sites showing a positive or a negative correlation? Did you analyse them separately? Are the sites showing decreased DNAm levels with your phenotype the same DNAm sites which are also negatively correlated with brain DNAm?
3. A validation dataset of Huntington's patients was used, however, this was not one of the diseases studied according to the main aim. Could you elaborate how Huntington brain DNAm could serve as validation?

Reviewer #2 (Remarks to the Author):

Mendonça et al. have investigated the potential use of blood-brain CpGs as biomarkers in neurodegenerative disease. This is an interesting topic, and the overall design of the study is sound. Demographics and description of the datasets used is well described in tables, and figures provided are useful in interpretation of study design and results. In general the authors have clearly explained the use of different datasets/types to investigate the research question, but there are points when it is difficult to follow. My suggestions for improvement are provided below:

Reviewer Comment 1: It would be beneficial for the authors to make clearer the use of blood/brain datasets and the diseases they correspond to, and to justify the use of these datasets.

Authors' answer:

The selection of diseases for our analysis was based on: (i) availability of datasets containing both neurodegenerative disease patients and health individuals, (ii) beta values obtained using the same methodologies employed to identify the bb-CpGs (Infinium HumanMethylation 450K and Infinium EPIC 850K) and (iii) diseases with brain permeability to peripheral cells linked to its biology.

Our hypothesis was that the methylation profiles of peripheral lymphocytes were modified after circulation in brain tissue from patients with specific neurodegenerative conditions. To clarify these points, we modified the Introduction and the Material and Methods (section "Establishment of the methylation profile in samples from individuals with neurodegenerative diseases"), to better justify our approach and detailing the criteria used to select the datasets and the diseases analyzed in this work. These sections were modified as follow:

i- Introduction:

"The use of blood cells as a surrogate for some neurodegenerative diseases can reflect the molecular and cellular changes involving various immunological mediators, including T lymphocytes. This type of cell accounts for part of nucleated blood cells and can cross the blood-cerebrospinal fluid barrier and the blood-brain barrier, circulating and being present in both brain tissue and blood (YAMAZAKI; KANEKIYO, 2017). The cross of these barriers is supported by the detection of T lymphocytes in brain tissue of Alzheimer's (CHEN et al., 2023), Parkinson (MCGEER et al., 1988; BROCHARD et al., 2009) and Multiple Sclerosis (PRAT et al., 2002) diseases. Since DNA methylation can be influenced by the cell environment, we hypothesized that blood cells can acquire and maintain specific CpG methylation patterns after circulating in the brain and upon its return to the bloodstream.

In this study, we looked for CpGs for which the level of methylation in blood DNA is correlated with the level observed in brain tissues in patients with neurodegenerative disorders. Our findings show that there is a distinct methylation profile for specific CpGs in individuals affected by Alzheimer's, Parkinson's, Multiple Sclerosis, and Fragile X Syndrome when compared with healthy subjects."

ii- Material and Methods (section "Establishment of the methylation profile in samples from individuals with neurodegenerative diseases"):

"To establish the model, we compared the beta-values for the 18,293 CpGs between patients and healthy controls. The beta-values, which represent the proportion of methylated CpG, were estimated by the ratio of methylated/(methylated+unmethylated) signals and were obtained from patients with neurodegenerative diseases in the Gene Expression Omnibus (GEO) database. In this study we adopted an exploratory approach, respecting the normalization and patient selection applied by the authors of each dataset, without specific sub-division based on age, sex, ethnicity, or disease stage. Only studies that evaluated the methylation profile using

the Infinium HumanMethylation450K or Infinium Methylation EPIC arrays were included, as these are the methodologies used in IMAGE-CpG and BECon. The neurodegenerative diseases used for comparison were defined based on the availability of blood datasets (Table 1) containing both patients and healthy individuals and included: Alzheimer's (GSE153712), Parkinson's (GSE111629), Multiple Sclerosis (GSE106648), and Fragile X Syndrome (GSE41273). The inclusion of Fragile X Syndrome is due to the association between Fragile X Mental Retardation Protein (FMRP) pathway and the translation of Amyloid Beta Precursor Protein (WESTMARK AND MALTER, 2007; RENOUX et al, 2014; and PIERGIORGE et al., 2023), whose accumulation in the brain is a hallmark of Alzheimer's Disease."

Reviewer Comment 2: There are many available brain DNA methylation datasets that correspond to different neurodegenerative diseases. It might be beneficial to compare results to a wider range of diseases/those with similar pathology. As the manuscript is reasonably short, there is room to expand the analysis. The datasets used for the Alzheimer's brain cohort are relatively old, it would be beneficial to look at DMPs from more recent studies.

Authors' answer:

As stated in the previous comment, we restricted our analyses to the datasets that employed the same strategy used by IMAGECpG and BeCon to identify the CpGs with correlated brain/blood methylation levels and presenting data from patients and health controls. In our search, the neurodegenerative diseases presented in the manuscript were those that fulfilled these criteria when the work was conceived in 2022.

However, searching for more recent datasets, following the reviewer's suggestion, we found an additional dataset for Parkinson Disease (GSE195834) that fulfills our criteria (deposited in April 20th, 2022), which compared methylation patterns of brain tissues between patients and healthy controls. The results from the analysis of this dataset were included in Results section ("**Other neurodegenerative diseases**"), and the text added was as follows:

"To carry out comparisons considering these nine bb-CpGs in brain tissues, the single data set meeting our criteria (i.e., analyzed by HumanMethylation 450K or Infinium EPIC 850K and containing patients and controls in the same dataset) was GSE195834, based on samples of primary motor cortex. Of the nine CpGs, eight presented the hypomethylated pattern found in blood samples, although only for one CpG the difference was statistically significant (cg12230709; Supp. Figure 2)."

Reviewer Comment 3: There are several grammar/spelling mistakes, it would be beneficial to review the manuscript to identify and rectify such instances.

Authors' answer:

We revised the text to correct the mistakes. All changes were highlighted in yellow.

Reviewer Comment 4: In results section 1 (Selection of CpG presenting correlation of DNA methylation levels between peripheral blood and brain), clarification on the numbers of CpGs identified would be beneficial. It is confusing to the reader about where the different number of CpGs referred to come from.

Authors' answer:

We modified the text trying to make clear how the CpGs were identified, and the first paragraph of this section was modified as follows:

“To identify the CpGs presenting correlated methylation levels between blood and brain tissues in paired samples, we analyzed the databases IMAGECpG and BeCon. First of all, 47,531 CpGs presented methylation levels with correlation coefficient $\geq |0.70|$, when comparing blood and brain-paired samples, being: 47,360 from the IMAGECpG dataset, and 1,131 from the BeCon dataset. IMAGECpG utilized two methylation arrays (HumanMethylation 450K and Infinium EPIC 850K), and CpGs present in both arrays exhibiting inconsistent results regarding the correlation between blood and brain were excluded. We identified 11,772 CpGs exclusive to the Infinium EPIC 850K array (considered the most comprehensive), and an additional 5,390 CpGs shared between the HumanMethylation450K and Infinium EPIC 850K arrays and with concordant results in both analyses. Furthermore, we included the 1,131 CpGs from the Becon dataset, resulting in a total of 18,293 CpGs (Figure 1 and Supp. Material 1). These CpGs are hereafter referred to as bb-CpGs, from blood and brain correlated CpGs.”

Reviewer Comment 5: Results section 2 (Establishment and validation of the blood-brain correlation model) is unclear. Consider rewording the results to explain more clearly what the purpose of the use of the different datasets in this section was.

Authors' answer:

We modified the text of this section trying to make it clearer. The objective of this section was to identify which CpGs (among the 18,293 presenting a correlation $\geq |0.70|$ in blood-brain tissue comparison) were differentially methylated in blood samples from patients with Alzheimer's Disease when compared with health controls, and to verify if these CpGs were also differentially methylated when comparing brain tissues (patients vs. health controls). This section was modified as follows:

“Establishment and validation of the blood-brain correlation model

To identify sites with differentially methylated levels in blood and to verify if the same profile could be found in the brain, the methylation of the bb-CpGs was analyzed in two case/control datasets of Alzheimer's disease: GSE153712, comprising 161 patients and 471 controls, for blood samples; and GSE72778, comprising 125 samples from Alzheimer's patients and 135 control samples, for brain tissues. A total of nine bb-CpGs were found to be differentially methylated, being all hypermethylated in blood samples of AD patients in comparison to healthy controls (Figure 3A). The chromosomes, genes, and locations of these bb-CpGs in respect to gene structure and CpG islands are shown in Table 2.

To evaluate if these nine bb-CpGs present a similar differential methylation profile between brain samples from AD patients and healthy controls, we used the GSE72778 dataset. All brain DNA samples were grouped for disease or control group, and tissue samples from different brain regions were analyzed together. The results showed that all nine bb-CpGs were also significantly hypermethylated in brain tissue of AD patients, similar to what was found in the analysis with DNA derived from blood samples (Figure 3B).”

Reviewer Comment 6: It should be acknowledged that there are limitations to the use of blood data as a surrogate, given that the number of CpGs with blood brain correlation is relatively small.

Authors' answer:

We agree with the reviewer that the number of CpGs with blood-brain correlation is small (~18,000). However, we believe that these CpGs could reflect the biological effect of these diseases in the bloodstream. Furthermore, the CpGs showing significant association with the diseases were mapped to (or next to) genes previously associated with the condition

(particularly Alzheimer's and Parkinson Disease). Anyhow, a sentence was added to the Discussion Section to highlight this limitation:

"Although analyzing DNA methylation in blood as a surrogate for brain has its limitations with regard to underrepresentation of the alterations that take place in the disease-affected tissue, it has a potential use in the identification of biomarkers that could help diagnosis and disease monitoring."

Reviewer #3

Mendonça and co-workers selected DNA methylation (DNAm) sites showing high correlation between blood and brain tissue based on previously datasets. These selected DNAm sites were tested for associations with several neurodegenerative diseases. In total, they identified 64 DNAm-disease associations. Mendonça and co-workers conclude that these findings support the rationale for using blood as a surrogate tissue for brain-related phenotypes.

My main concerns are about the validity of the DNAm association analysis. I Miss a clear description on pre-processing of the DNAm data and the linear regression models used for determining the associations.

Authors' answer:

We thank the reviewer for the comment. Since we used different datasets and the available information for each of them also differed, including availability of raw data and patients' characteristics, we chose to base all our analyses on beta-values, provided in all datasets. These values refer to the ratio between the intensity of the methylated allele and the sum of intensities of the methylated and unmethylated alleles. Therefore, the pre-processing was not performed by us and followed the criteria established in each study. Since this could represent a bias, we did not perform comparisons between studies, i.e. the analyses were performed individually within each dataset (comparisons between cases and controls). Regarding the correlation analysis between blood and brain methylation levels, this was also performed within the same dataset, guaranteeing that the pre-processing for the different tissues was performed following the same criteria. To further strengthen our analysis, we focused on strong correlations ($r > |0.7|$), and all linear regression models were adjusted for multiple comparisons (Benjamini-Hochberg).

Major comments:

Introduction

1. You mention that DNAm from surrogate tissues (i.e., lymphocytes) are used to inform on diagnoses and prognoses of various diseases. The diseases you mention are all peripheral diseases, and one could argue that these peripheral diseases have widespread effects on the peripheral epigenome through e.g., the immune system. Your next conclusion is the importance to study peripheral DNAm in brain-related disorders. Could you provide a better rational why you think that peripheral DNAm might be informative for brain, specially given the presence of blood-brain barrier (BBB)?

Authors' answer:

Thank you for your comment. We tried to make our rational more clear by modifying the Introduction section providing additional support for the use of other tissue samples as surrogates from brain tissue [references Sun et al (2023), Martínez-Iglesias et al (2021)] and for the presence of peripheral lymphocytes in brain tissue in the selected neurodegenerative diseases [references CHEN et al. (2023), MCGEER et al. (1988), BROCHARD et al.(2009) and PRAT

et al.(2002)]. We also modified the second paragraph of the Introduction section to support the use of blood cells and to better detail the objective of the manuscript. This part of Introduction was modified as follows:

“The use of blood cells as a surrogate for some neurodegenerative diseases can reflect the molecular and cellular changes involving various immunological mediators, including T lymphocytes. This type of cell accounts for part of nucleated blood cells and can cross the blood-cerebrospinal fluid barrier and the blood-brain barrier, circulating and being present in both brain tissue and blood (YAMAZAKI; KANEKIYO, 2017). The cross of these barriers is supported by the detection of T lymphocytes in brain tissue of Alzheimer’s (CHEN et al., 2023), Parkinson (MCGEER et al., 1988; BROCHARD et al., 2009) and Multiple Sclerosis (PRAT et al., 2002) diseases. Since DNA methylation can be influenced by the cell environment, we hypothesized that blood cells can acquire and maintain specific CpG methylation patterns after circulating in the brain and upon its return to the bloodstream.

In this study, we looked for CpGs for which the level of methylation in blood DNA is correlated with the level observed in brain tissues in patients with neurodegenerative disorders. Our findings show that there is a distinct methylation profile for specific CpGs in individuals affected by Alzheimer’s, Parkinson’s, Multiple Sclerosis, and Fragile X Syndrome when compared with healthy subjects.”

2. The introduction lacks a clear rational why it is important to study the epigenome in neurodegenerative disorders. Could you expand on the study rational? Could you also expand on why you specially study AD, PD, MS, and Fragile X?

Authors’ answer:

We modified the Introduction providing additional support for the rational of the use of DNA methylation alterations as potential biomarkers for neurodegenerative diseases, citing authors that support or carried out similar approaches. Please see our answer in the comment immediately above. In respect to the diseases analyzed, our interest in AD, PD and MS was due to the presence of lymphocytes in brain tissues of patients affected by these conditions, as supported by CHEN et al. (2023), MCGEER et al. (1988), BROCHARD et al.(2009) and PRAT et al. (2002). In respect to fragile X, the inclusion was due to “the association between Fragile X Mental Retardation Protein (FMRP) pathway and the translation of Amyloid Beta Precursor Protein (WESTMARK AND MALTER, 2007; RENOUX et al, 2014; and PIERGIORGE et al., 2023)”, as mentioned in Material and Methods section. An additional point is the availability, in public databases, of methylation data for these conditions (AD, PD, MS, and Fragile X), obeying the criteria used for data selection.

3. “The use of blood cells as a surrogate for some neurodegenerative diseases can reflect the molecular and cellular changes involving various immunological mediators, including T lymphocytes.” Could you provide references for these statements?

Authors’ answer:

We provided these references in introduction section. Please, see the answers for the comments above.

4. “This type of cell accounts for about ~70% of nucleated blood cells and can cross the blood-cerebrospinal fluid barrier and the blood-brain cells and can cross the blood-cerebrospinal

fluid barrier and the blood-brain barriers, circulation and being present in both brain and blood (YAMAZAKY; KANEKIYO, 2017).” The study cited here, discussed the possibility of a leaky BBB in Alzheimer’s disease specifically. How does this translate to the other diseases you investigate in your manuscript? Can you assume this is the case for every neurodegenerative disorder, and why?

Authors’ answer:

Thank you for your comment. We do not think that this is a characteristic of every neurodegenerative disease. However, we believe that the presence of lymphocytes in the brain can be facilitated by specific conditions. As mentioned above, this was supported by the publications cited in the text concerning AD, PD and MS.

Methods

1. “The CpGs indicated by at least one database with a positive or a negative correlation above 70% were selected, respecting the statistical analysis used by each of them.” Could you clarify whether you are using the correlation coefficient of $|0.7|$, or the explained variance of 70%?

Authors’ answer:

Thank you for highlighting this point. We clarified this point in Material and Methods section, item “Selection of CpG sites with correlated DNA methylation pattern between peripheral blood and brain”. In fact, we were referring to a correlation coefficient $> |0.70|$.

2. How did you deal with the DNAm sites showing a positive or a negative correlation? Did you analyse them separately? Are the sites showing decrease DNAm levels with your phenotype the same DNAm sites with are also negatively correlation with brain DNAm?

Authors’ response:

We considered the correlation values as absolute values ($> |0.70|$). Consequently, we did not analyze separately the positive or negative correlations. In respect to changes in methylation level, we provided this information for the differentially methylated CpGs for each cases-controls comparison, as presented in Tables 1-5.

3. A validation dataset of Huntington’s patients was used, however, this was not one of the diseases studied according to the main aim. Could you elaborate how Huntington brain DNAm could serve a validation?

Authors’ answer:

Thank you for your comment. We agree that Huntington was in disaccord. We decided to exclude the analysis carried out for Huntington Disease. In the previous version of the manuscript, this analysis compared the methylation levels in brain tissues between patients with Huntington and health controls for the nine CpGs identified in AD patients. Consequently, the results section “Comparison with other neurodegenerative diseases: brain samples”, that reported these findings, was excluded.

Additional changes made in the manuscript.

- The Supporting Figure 1 was deleted in the revised version, the data corresponding to the comparison between Alzheimer’s Disease patients and healthy controls, showed in this figure, are now presented in Figure 3b (as a Volcano Plot).

-The following references were added to the manuscript:

Alisch R.S *et al.* Genome-wide analysis validates aberrant methylation in fragile X syndrome is specific to the FMR1 locus. *BMC Med Genet.* 14, 18; 10.1186/1471-2350-14-18 (2013).

Blanco-Luquin, I. *et al.* Early epigenetic changes of Alzheimer's disease in the human hippocampus. *Epigenetics.* 15, 10, 1083–1092; 10.1080/15592294.2020.1748917 (2020).

Brochard, V. *et al.* Infiltration of CD4+ lymphocytes into the brain contributes to neurodegeneration in a mouse model of Parkinson disease. *J Clin Investv.* 119, 1, 182–192, 10.1172/JCI36470 (2008).

Chuang Y. H. *et al.* Parkinson's disease is associated with DNA methylation levels in human blood and saliva. *Genome Med.* 9,1, 76; 10.1186/s13073-017-0466-5 (2017).

Horvath, S. *et al.* Huntington's disease accelerates epigenetic aging of human brain and disrupts DNA methylation levels. *Aging.* 8, 7, 1485-1512. 10.18632/aging.101005 (2016).

Kular L, Liu Y, Ruhmann S, Zheleznyakova G *et al.* DNA methylation as a mediator of HLA-DRB1*15:01 and a protective variant in multiple sclerosis. *Nat Commun.* 9,1,2397; 10.1038/s41467-018-04732-5 (2018).

Martínez-Iglesias O. *et al.* Epigenetic Biomarkers as Diagnostic Tools for Neurodegenerative Disorders. *Int J Mol Sci,* 23, 1; 10.3390/ijms23010013 (2022).

McGeer, P. *et al.* Rate of cell death in parkinsonism indicates active neuropathological process. *Ann neuropol.* 24, 4, 574-576; 10.1002/ana.410240415 (1988)

Nabais M. F. *et al.* Meta-analysis of genome-wide DNA methylation identifies shared associations across neurodegenerative disorders. *Genome Biol.* 22, 1, 90; 0.1186/s13059-021-02275-5. (2021)

Prat A. *et al.* Migration of multiple sclerosis lymphocytes through brain endothelium. *Arch Neurol,* 59,3, 391-397; 10.1001/archneur (2002).

Piergiorgio R.M. *et al.* Multi-layered transcriptomic analysis reveals a pivotal role of FMR1 and other developmental genes in Alzheimer's disease-associated brain ceRNA network. *Comput Biol Med.* 166, 107494; 10.1016/j.compbio.2023.107494 (2023).

Renoux, A. J. *et al.* Fragile X mental retardation protein expression in Alzheimer's disease. *Frontiers in genetics,* 5, 360; 10.3389/fgene.2014.00360 (2014)

Steger, M. *et al.* Phosphoproteomics reveals that Parkinson's disease kinase LRRK2 regulates a subset of Rab GTPases. *eLife,* 5, 10.7554/ELIFE.12813 (2016).

Sun, Y. *et al.* Identification of candidate DNA methylation biomarkers related to Alzheimer's disease risk by integrating genome and blood methylome data. *Transl Psychiatry,* 13, 387; 10.1038/s41398-023-02695-w (2023).

Vishweswaraiah S. *et al.* Methylated Cytochrome P450 and the Solute Carrier Family of Genes Correlate With Perturbations in Bile Acid Metabolism in Parkinson's Disease. *Front Neurosci,* 16,804261; 10.3389/fnins.2022.804261 (2022).

Westmark, C. J, & Malter J. S. FMRP mediates mGluR5-dependent translation of amyloid precursor protein. *PLoS Biol.,* 5, 3; 10.1371/journal.pbio.0050052 (2007).

REVIEWERS' COMMENTS:

Reviewer #2 (Remarks to the Author):

The authors have addressed the concerns laid out in the first review, I believe the report is now ready for publication.